# Therapeutic Evaluation *Punica granatum* Peel Powder for the Ailment of Inflammatory Bowel Disorder in NCM460 Cell Line and in Albino Rats

**DOI:** 10.3390/pharmaceutics17070843

**Published:** 2025-06-27

**Authors:** Parikshit Roychowdhury, Gyanendra Kumar Prajapati, Rupesh Singh, Prasanna Gurunath, Ramesh C, Gowthamarajan Kuppuswamy, Anindita De

**Affiliations:** 1Department of Pharmaceutics, JSS College of Pharmacy, JSS Academy of Higher Education and Research, Ooty 643001, Tamil Nadu, India; parikshitroychowdhury@gmail.com; 2Department of Pharmacology, KLE College of Pharmacy, KLE Academy of Higher Education and Research, Belagavi 590010, Karnataka, India; gyanendra111096@gmail.com; 3Department of Pharmacology, St. John’s Pharmacy College, Vijayanagar, Bengaluru 560104, Karnataka, India; singhrupesh272@gmail.com; 4Department of Pharmacology, SBD Institute of Pharmacy, Bengaluru 560019, Karnataka, India; prasannages@gmail.com; 5Department of Pharmacology, East West College of Pharmacy, Bengaluru 560091, Karnataka, India; rameshcology80@gmail.com; 6Department of Pharmaceutics, College of Pharmacy, JSS University, Noida 201301, Uttar Pradesh, India; 7Department of Pharmaceutics, College of Pharmacy, Ajou University, Suwon 16499, Republic of Korea

**Keywords:** *Punica granatum*, inflammatory bowel disease, pre-clinical study, enterocolitis, chemical induced colitis

## Abstract

**Background:** Ulcerative colitis (UC) is a chronic inflammatory condition associated with the colon and rectum, often predisposing individuals to inflammatory bowel disease-related colorectal cancer (IBD-CRC). Current therapeutic options for UC, including corticosteroids and immunosuppressive drugs, pose significant side effects. *Punica granatum* peel powder (PPPG), a traditional herbal remedy in Ayurveda medicine for colitis, exhibits promising therapeutic effects with a favorable safety profile. **Objectives:** This study aims to explore the therapeutic potential and mechanism of action of a modified PPPG formulation in UC treatment. **Methods:** Using NCM460 cells and an acetic acid-induced UC murine model, the efficacy of modified PPPG was evaluated. **Results:** Therapy with modified PPPG significantly improved UC-associated symptoms, such as improvements in body weight, colon length, and disease activity index, as validated by histological examination. Transcriptomic sequencing identified downregulation of the IL-6/STAT3 signaling pathway and reduced inflammatory markers like p-NF-κB, IL-1β, and NLRP3 on PPPG therapy. **Conclusions:** These findings suggest that modified PPPG holds promise as a novel therapeutic strategy for UC intervention, targeting key inflammatory pathways implicated in UC pathogenesis and potentially mitigating the risk of IBD-CRC.

## 1. Introduction

Inflammatory bowel diseases (IBDs) represent a well-recognized group of chronic, relapsing conditions characterized by an intricate interplay of genetic predisposition, altered immune responses, and breaks in the gut microbiome. Among these conditions, Crohn’s disease (CD) and ulcerative colitis (UC) have garnered significant attention due to their complex nature and propensity for relapse [1]. With an estimated 5 million cases in 2023 for UC and 4.9 million cases in 2019 of IBD, it poses a global concern; there has been a 47.5% increase in IBD occurrence since 1990. Among all regions, UC and Crohn’s disease both have the highest occurrence in Europe, with 505 and 322 patients per 100,000 population; second to it is the USA. This statistic aligns with that of colorectal cancer as per GLOBOCAN 2024 as well, indicating a higher lower GIT issue in Western countries. In UC, the inflammation is predominantly confined to the colon and rectum, presenting as superficial inflammation. On the other hand, CD involves the entire gastrointestinal tract, resulting in ulceration, hemorrhage, and edema. Notably, the inflammation in CD extends through all layers of the intestinal wall, creating a transmural pattern. UC is characterized by inflammation at the mucosal level, leading to discernible anatomical changes, and in about 25% of cases, the inflammation can extend up to the rectum. This leads to ulceration, edema, and profuse bleeding. Histologically, both acute and chronic phases of UC exhibit mucosal inflammation marked by the presence of mononuclear and polymorphonuclear neutrophils [2]. In contrast, CD can affect any part of the GIT, from the oropharynx to the perianal region. The affected areas are often discontinuous, resulting in what is termed “skip areas” [3]. The inflammation associated with CD is more complex, involving the formation of fistulas and extending transmurally through the serosa. Superficial ulcerations, particularly on Peyer’s patches, are a hallmark, accompanied by chronic inflammation that can reach the submucosa and give rise to non-caseating granulomas [4]. The clinical presentation of IBD includes symptoms such as abdominal pain, cramping, diarrhea, dysentery, constipation, fatigue, and fever. These diseases impose a substantial burden on healthcare systems and socioeconomic status due to their frequent relapses and exacerbations, leading to a decrease in Quality Adjusted Life Years (QALYs) [5,6].

While both conditions are characterized by recurring and incurable natures, UC is thought to arise from intricate interactions between genetic factors, environmental influences, and dysregulation of the gut microbiome. Management of IBD revolves around anti-inflammatory and immune-modulatory interventions, including medications like sulfasalazine, corticosteroids, methotrexate, and infliximab (an anti-TNF-α drug) [7]. Additionally, antibiotics and probiotics find utility in addressing infections, promoting fistula healing, and restoring the balance of the gut microbiome. Apart from their high cost, these treatments impose serious shortcomings, such as a 30–40% loss of response with a prolonged treatment period via anti-drug antibodies for biologics, and also cause adverse reactions like increased likelihood of tuberculosis reactivation, non-melanoma skin cancers, and lymphoma where the immunosuppressants such as methotrexate and azathioprine cause chronic immunodeficiency, leading to susceptibility to fungal and systemic infections as well as risk of malignancy. Corticosteroids, on the other hand, can cause side effects like mood swings, osteoporosis, diabetes, etc. [8]. An extensive examination of the existing literature underscores a range of factors that contribute to the development of IBD. These encompass genetic predisposition, environmental triggers, the intricate balance of the gastrointestinal microbiota, early-life exposure to antibiotics, personal hygiene practices, infectious agents, antibiotic usage, oral contraceptive use, non-steroidal anti-inflammatory drug consumption, lifestyle choices, dietary habits, and stress levels [9,10]. A comprehensive understanding of these multifaceted factors holds the key to refining the management and prevention of IBD. Herbal products have consistently left a profound impact on the management of chronic disorders, often serving as the foundation for many modern medicines. The region of IBD management is no exception, as herbal remedies have shown their potential. Among these, *Punica granatum* L. from the Lythraceae family has emerged as a notable challenger, demonstrating effectiveness as an herbal remedy across various inflammatory conditions [11] through its targeted action on the IL-6/STAT3 pathway, a key player in inflammation. Noteworthy for its application in both Ayurveda and Unani medicine, the polyphenolic extract of *Punica granatum* L. has attracted significant interest in the Western world [12]. This interest primarily revolves around its potential in treating and preventing inflammatory diseases such as arthritis. *Punica granatum* and its active constituents have exhibited a multitude of therapeutic effects, spanning from anti-inflammatory and antioxidant properties to gastroprotective, immunomodulatory, antinociceptive, antimicrobial, antidiabetic, and even anticancer activities [13]. While prior investigations have predominantly centered on the juice extract of *Punica granatum*, our recent research has unveiled the significant inflammatory potential residing in the peel extract of the fruit, known as *Punica granatum* peel extract (PPPG). This extract, rich in bioactive compounds, has demonstrated comparable therapeutic efficacy for managing UC and IBD. The current research focused on the careful examination of the decoction of PPPG for its therapeutic effectiveness in a murine model of colitis induced by acetic acid. The research tests the mechanism of action underlying its impact on the IL-6/STAT3 inflammatory signaling pathway (Figure 1). This pathway plays a pivotal role in the initiation and progression of inflammation, and understanding how PPPG modulates it is crucial for comprehending its potential in alleviating UC.

## 2. Materials and Method

### 2.1. Peel Powder of Punica Granatum (PPPG) Sample Preparation

*Punica granatum* fruits were carefully sourced from the local market of Bangalore, Karnataka, India, locally known as K R Market, with the GPS latitude and longitude of 12.96592, 77.57611. There are no known large industries, natural reserves, or major state or national highways nearby the collection location; therefore, it was assumed that the peels were free from chemical and polluting entities. Only intact, fresh fruits without any alterations or damage on the peel were chosen for the study. The goal was to ensure the use of whole, unaltered fruits. To extract the peels, the fruits were selected and subjected to a drying process. The peels were meticulously separated and then subjected to drying in a hot air oven (Marconi, Piracicaba, SP, Brazil) at a controlled temperature of 45 °C for a duration of 48 h. Once the peels were sufficiently dried, they were subjected to a shade-drying process to maintain their integrity. The dried peels were then transformed into a coarse powder (using a sieve with a mesh size of 60) through the use of a grinding blender (Philips HL7756 model with a power consumption of 750 watts) [14]. To ensure the removal of wax components, the coarsely powdered material underwent a dewaxing process using n-hexane. The subsequent steps involved sequential extraction of the powdered material using various solvents in a Soxhlet extraction assembly. Ethyl acetate, methanol, and ethanol were employed for extraction in a stepwise manner.

Before every extraction with a new solvent, the powdered sample was pre-dried in a hot air oven below 50 °C [15]. A water extract was achieved using a cold maceration procedure in a closed vessel. Each solvent extraction was then concentrated by distillation of the solvents. Each extract’s yield was determined by weighing the final dried extract and expressing the percentage yield based on the original plant material weight. All extraction fractions were carried out to assess the phytochemical components present in the peel extracts for future and upcoming studies. Our study used the whole peel powder in a suspended form with water for injection.

### 2.2. Preliminary Phytochemical Analysis

Each extract fraction of PPPG was subjected to a series of qualitative chemical tests for the identification of its plant constituents, as per the protocol laid down in JB Harborne’s reference [16,17]. To more clearly determine its composition, the ethyl acetate, methanol, ethanol, and aqueous extracts were examined through thin-layer chromatography (TLC) on silica gel plates (Merk, Darmstadt, Germany) as adsorbent and water–acetonitrile (7:3 *v*/*v*) as the mobile phase. Different spots that appeared in each of the solvent systems were characterized using a range of techniques, including an iodine chamber, ultraviolet (UV) light (365 nm), and alcoholic KOH (SDFCL Ltd., Mumbai, India) solution (5%). Rf (retention factor) values for each spot were derived to assess the distribution of the isolated chemicals. Based on the Rf values and distinct appearances of spots under different visualization techniques, the optimum solvent system for effecting separation of constituents was evaluated.

This process of qualitative chemical testing and TLC analysis provided valuable insights into the composition of the PPPG extract, aiding in the identification of its individual plant constituents. Chlorogenic acid and rutin (Sigma Aldrich^®^, St. Louis, MO, USA) were procured as reference standards with purity of ≥95%.

### 2.3. Estimation of Total Phenolic Content

Total phenolics were determined by the Folin–Ciocalteu method of Dewanto et al. (2002) [18]. 100 µL of each extract was carefully pipetted into a 25 mL volumetric flask. 10 mL of water and 1.5 mL of Folin–Ciocalteu reagent (Alpha Chemika, Mumbai, India) were added. The reaction mixture was left to stand for 5 min before adding 4 mL of a 20% *w*/*v* sodium carbonate (SDFCL Ltd., Mumbai, India) solution. The level of the solution was adjusted to 25 mL using double-distilled water [19]. The resultant solution was then allowed to stand for approximately 30 min; gradually a clear blue coloration began to appear. For the establishment of phenolic content, the solution was quantified at 765 nm using a UV–visible spectrometer (Shimadzu, UV-1601, Kyoto, Japan). Gallic acid (Yucca Ltd., Bangalore, India) was used as a standard reference for calculating the percentage of total phenolic content in the extract.

### 2.4. Animal Experiment

Experimental animals Albino Wistar rats weighing between 180 and 200 g needed for the experiments were obtained from commercial CPCSEA-approved animal house in Ben-galuru and were quarantined at the animal house for a week prior to use. All the animals were kept at 22–24 °C temperature, under 12:12 h light–dark cycle and were given ad libitum (Nutricubo, Purina, Torrance, CA, USA) feed and water under CPCSEA-approved animal house facility. Animals were allocated randomly in groups of seven and acclimatized in the housing 5 days before the start of the experimental protocol.

### 2.5. Primary Selection of Extraction for the Effective Anti-Inflammatory Activity in HRBC Membrane Stabilization Method

The anti-inflammatory activity of various extracts of PPPG was analyzed by an in vitro approach through human red blood cell (HRBC) membrane stabilization [20]. The blood collected from the blood bank was diluted with an equal volume of Alsever’s solution that is comprised of dextrose (Nice Chem Ltd., Kerala, India) (2%), sodium citrate (SDFCL Ltd., India) (0.8%), citric acid (SDFCL Ltd.) (0.05%), sodium chloride (SDFCL Ltd.) (0.42%), and distilled water (100 mL). The above mixture was then centrifuged in an iso-saline solution. Remi C-30 plus cooling centrifuge with R-240 AM rotor head was used for this and all the following studies. Subsequently, 1 mL of the HRBC suspension was mixed with an equivalent volume of the test drug in three varying concentrations. The assay mixtures were incubated at 37 °C for 30 min and subsequently centrifuged again at 3000 rpm for 20 min. Hemoglobin concentration in the supernatant solution was determined by spectrophotometer, and the reading was taken at a wavelength of 560 nm. Diclofenacsodium usage as regular drug. To find the percentage of haemolysis, the above formula was applied:$$\%\;Haemolysis\;=\;\frac{Haemoglobin\;content\;in\;test\;sample}{Haemoglobin\;content\;in\;total\;lysed\;cells}\;\times 100$$

Conversely, % HRBC membrane stabilization calculated by following formula.% Protection = 100 − % Haemolysis

### 2.6. Cell Culture for the Anti-Inflammatory Activity

The colon epithelium cell line NCM460 was procured from INCELL Corporation and stored in the Korean cell bank (with the code NCM460) (cell bank, Seoul, Republic of Korea) and cultured at 37 °C using Dulbecco’s Modified Eagle Medium (DMEM) (Himedia, Kennett Square, PA, USA) with 10% fetal bovine serum (FBS) and 1% penicillin and streptomycin (P/S) solution at 5% CO_2_ humidified condition for 5 days [21].

#### 2.6.1. Cellular Toxicity Study

NCM460 cells seeded in a 96-well plate and treated with 1–100 µg/mL concentration of PPPG for 24 and 48 h and further incubated with 75 mg/mL of acetic acid. The EZ-Cytox Cyto-toxicity Test Kit was utilized to determine the cytotoxicity of cells through the 3- (4,5-dimethylthiazol-2-yl)-2,5-diphenyltetrazolium bromide (MTT) assay (DoGen Bio, Seoul, Republic of Korea). DMSO with a concentration of 0.1% (*v*/*v*) was used as control. The DMSO A small quantity of DMSO was added in the in vitro assays (Section 2.6.1) as the solvent only for the pur-pose of solubilization of some poorly water-soluble phytochemicals in the Punica gran-atum peel extract. The concentration of DMSO in the culture medium was maintained be-low 0.1% to ensure that it would have no cytotoxic effects against NCM460 cells. The media was replaced after 24 and 48 h, and cells rinsed twice with 1 mL of PBS. Then, 20 µL of MTT solution (stock: 5.0 mg/mL in PBS) [22] was applied to each well. The cells were incubated for 4 h to allow the activation of mitochondrial dehydrogenases. Finally, formazan absorbance at 570 nm was estimated by using a microplate reader (VIC-TORTM X3, PerkinElmer). The experiment was done three times [23].

#### 2.6.2. Western Blot Analysis

Western blotting (ChemiDoc MP Imaging System, Bio-Rad Laboratories, Hercules, CA, USA) experiments were done according to our earlier procedure [24,25]. NCM460 cells were cultured in 6-well plates and further incubated for 24 h using different concentrations of PPPG. Upon incubation, radioimmunoprecipitation (RIPA) buffer was used to lyse the cells. The cells were centrifuged at 4 °C (12,000× *g*, 30 min), and the supernatants were collected. The collected material was electrophoresed for the proteins. The concentration of total proteins was measured by Bradford’s method. Proteins (30 g/lane) were then run on 10% SDS-PAGE and subsequently blotted onto PVDF membranes. The membranes were then incubated with primary antibodies IKK (clone D31C6), IκB-α (Clone: L35A5), *p*-Iκb-α (Clone: 14D4), NF-κB (Clone: D14E12), Nrf2 (Clone: D1Z9C), and Keap-1 (Clone: D1J9M) (manufacturer of all the primary antibody was Cell Signaling Technology (Danvers, MA, USA)) overnight at 4 °C. Subsequently, the membrane was treated with horseradish peroxidase (HRP, GenDEPOT, Seoul, Republic of Korea)-conjugated secondary antibody at 1:3000 *v*/*v* for 2 h. Blots were finally developed using the enhanced chemiluminescence (ECL) kit. ImageJ software 2020 (version 1.53) was employed to detect each band, and all the results were represented as relative ratios to the corresponding phosphorylation site of glyceraldehyde-3-phosphate dehydrogenase (GAPDH, GenDEPOT, Gunpo, Seoul, Republic of Korea).

### 2.7. Induction of Colitis by Acetic Acid and Experimental Design

Rats were subjected to a 16–18 h fast and ad libitum access to water prior to the induction of colitis. Acute colitis was induced through a modified protocol described by Murrat et al. [26]. In short, healthy animals received rectal instillation of 2 mL/day of 4% acetic acid in three divided doses over consecutive days. Acetic acid (SDFCL Ltd.) was given through a lubricated plastic catheter, positioned to avoid spillage or outflow. The administration was done with a polyethylene catheter (PE-90) placed 8 cm proximal to the anus and a 90 s inversion period to avoid loss of the given solution.

Rats were randomly divided into seven groups (*n* = 5 each) as follows:(a)Control group: Received the same experimental handling as the test group but without acetic acid treatment. Drug treatment was substituted with vehicle (water for injection) administration, orally (5 mL/kg).(b)PPPG group: Treated orally or i.p. with PPPG (100 mg/kg) without acetic acid treatment, this group was maintained to observe any untoward effect or other than therapeutic effect of the peel powder.(c)Acetic acid treatment group: Received oral (5 mL/kg) or i.p. (2 mL/kg) vehicle after induction of colitis with acetic acid.(d)Acetic acid treatment plus low-dose PPPG group: Treated orally or i.p. with 3 mg/kg PPPG after acetic acid treatment.(e)Acetic acid treatment plus moderate-dose PPPG group: Treated orally or i.p. with 30 mg/kg PPPG after acetic acid treatment.(f)Acetic acid treatment plus high-dose PPPG group: Treated orally or i.p. with 100 mg/kg PPPG after acetic acid treatment.(g)Reference group: Treated intrarectally with 100 mg/kg 5-aminosalicylic acid (5-ASA) as a standard treatment after acetic acid treatment.

All treatments commenced 6 h post-acetic acid-induced colitis and were administered daily for four consecutive days. Animals were euthanized via pentobarbital overdose (200 mg/kg) four days after acetic acid treatment administration.

Body weight and average feed intake of experimental animals were recorded on day 0, day 8, and day 15.

Ulcer scoring was conducted on the inflamed colon of all test animals following the method outlined by Morris et al. [21]. The distal 8 cm of the colon were longitudinally opened, and the luminal contents were carefully cleared, rinsed with saline, and subsequently dried on filter paper. Macroscopic scoring of colonic lesions was performed, which outlines the scoring system for evaluating mucosal lesions [17,18]. This scoring system assigns scores based on the appearance of the lesions, ranging from 0 (indicating no damage) to 10 (indicating extensive damage extending more than 2 cm along the length of the colon), allowing for a comprehensive assessment of colonic tissue integrity and inflammation severity.

### 2.8. Biomarker Estimation

#### 2.8.1. Estimation of Myeloperoxidase (MPO)

The myeloperoxidase (MPO) activity in excised inflamed colon tissue was measured following a rigorous protocol. Firstly, the tissue samples were homogenized using ice-cold phosphate buffer. The resulting homogenate was then centrifuged, and the supernatant was collected. Subsequently, the supernatant was mixed with O-phenylenediamine (Sigma Aldrich Ltd., St. Louis, MO, USA) to initiate the enzymatic reaction. The absorbance of the reaction mixture at 492 nm was recorded at intervals of 30 s for a duration of 5 min using a spectrophotometer. The MPO activity was quantified using a calculation based on the change in absorbance per minute per unit volume of the reaction mixture. Specifically, one unit of MPO activity was defined as the change in absorbance per minute by 1.0 at room temperature. This activity was determined using the following formula [22], which accounts for the rate of change in absorbance over time and the volume of the reaction mixture.$$MPO\;Activity\;(\frac{U}{g})\;\frac{X}{weight\;of\;tissue\;piece}$$$$\mathrm{where}\;X\;=\;10\;\times \;\frac{change\;in\;absorbamce\;per\;minute}{volume\;of\;supernatent\;taken\;in\;the\;final\;reaction}$$

#### 2.8.2. Estimation of Lipid Peroxidation (Malondialdehyde) (MDA) in the Inflamed Colon

The method for assessing protein content in colonic tissue homogenate was adapted from Pomory et al. [23]. Initially, 0.5 mL of the homogenized tissue sample was mixed with 5 mL of copper tartrate bicarbonate solution and allowed to incubate for 10 min. Following this incubation period, 0.5 mL of 1N Folin Ciocalteu’s reagent (FCR) was added to the mixture, and the solution was left to stand for an additional 2 h. Absorbance readings were then obtained at 660 nm using a spectrophotometer, with a blank containing 1N NaOH solution and a standard consisting of bovine serum (10 mg/mL).

The second part of the assessment involved quantifying malondialdehyde (MDA) levels in colonic tissue, following the protocol described by Ohkawa et al. [24]. To begin, 0.5 mL of colonic homogenate was mixed with 2.5 mL of 10% trichloroacetic acid and then boiled on a water bath for 15 min. After boiling, the mixture was cooled and centrifuged at 3000 rpm for 10 min. From the resulting supernatant, 2 mL was collected and mixed with 1 mL of 0.67% thiobarbituric acid solution. This mixture was boiled for an additional 15 min, allowed to cool, and then the absorbance was measured at 532 nm against a blank containing the mixture without the sample. The MDA concentration was calculated using the following formula:$$MDA\;(nmol\cdot {mg}^{-1})\cdot Protein\;\frac{OD\;of\;T\;\times \;total\;volume\;of\;reaction\;mixture}{1.56\;\times \;10^{5\;}\times \;10^{-9}\;\times \;sample\;volume\;(mg\;protein/mL)}$$

-OD of T represents the optical density of the sample at 532 nm;-Total volume of the reaction mixture is the total volume of the reaction mixture in mL;-The nanomolar extinction coefficient of MDA (1.56 × 10^5^ M^−1^ cm^−1^) is used for conversion;-Sample volume (mg protein/mL) refers to the volume of the sample containing protein in mg per mL.

### 2.9. Histopathological Study: Excised Colon of Representative Sample from Various Roups Were Observed After H&E Staining by a Person Blind to the Treatment Protocol

Colon specimens from representative samples across various experimental groups were subjected to histological examination following hematoxylin and eosin (H&E) staining by an observer blinded to the treatment protocol. Tissue processing involved fixation of rat colon tissue in 10% formalin for 24 h to ensure optimal preservation. Subsequently, tissue sections were prepared through sequential steps including material extraction, dehydration, wax dipping, embedding, sectioning, H&E staining, and cover slipping.

The stained tissue sections were then scanned using a digital scanning system, and the electronic images were stored and analyzed on a computer. Histological damage was evaluated using a specific scoring method, whereby the pathological morphological features observed in each visual field were assessed and assigned corresponding scores as follows:-Score of 0: normal intestinal mucosa observed in the visual field.-Score of 1: mild inflammation and edema of the mucosal layer with disappearance of 1/3 of the crypts at the basal part.-Score of 2: moderate inflammation of the mucosal layer with disappearance of 2/3 of the crypts at the basal part.-Score of 3: moderate inflammation of the mucosal layer with complete disappearance of crypts, while the epithelial layer remains intact.-Score of 4: severe inflammation involving the mucosa, submucosa, and myometrium, with complete disappearance of crypts and epithelium.

This detailed scoring system enables precise quantification of histological damage severity and provides valuable insights into the pathological changes occurring in the colon tissues of experimental animals.

### 2.10. Statistical Analysis

The results were presented as Mean ± Standard Error of Mean (SEM) and analyzed using one-way analysis of variance (ANOVA), followed by post hoc Tukey’s comparison tests. All comparisons were made against both the non-colitis group of animals and the untreated colitis control group of animals to assess significance. Statistical significance was determined with a threshold of *p* ≤ 0.05. Data analysis was conducted using Prism8 software.

## 3. Result

The extractive values of ethyl acetate, methanol, ethanol, and water extracts were determined to be 4.09% *w*/*w*, 6.76% *w*/*w*, 14.4% *w*/*w*, and 19.70% *w*/*w*, respectively. These values provide insights into the efficiency of the extraction process and the yield of bioactive compounds from the peel of *P. granatum*. Preliminary phytochemical screening of these extracts indicated the presence of essential phytoconstituents such as carbohydrates, glycosides, saponins, tannins, and flavonoids, as summarized in Table 1. This analysis highlights the diverse chemical composition of the extracts and suggests their potential therapeutic benefits. Thin-layer chromatography (TLC) analysis was conducted on various extracts of *P. granatum* peel using different solvent systems (1–5) to assess the presence of solutes. Notably, the solvent system composed of benzene, acetic acid, and water in the ratio of 125:72:3 demonstrated superior separation efficiency, yielding a higher number of compounds compared with other solvent systems. Rf values were calculated to characterize the mobility of the solutes within the chromatographic matrix. Furthermore, the total phenolic content of the extracts was quantified using the Folin–Ciocalteu method, with gallic acid serving as the reference standard. The determined phenolic content was 195 µg/mL for water extract, 220 µg/mL for ethanol extract, 310 µg/mL for ethyl acetate extract, and 495 µg/mL for methanol extract. This assessment revealed variations in phenolic content among the extracts, with methanol extract exhibiting the highest phenolic content, followed by ethyl acetate, ethanol, and water extracts. This information elucidates the potential antioxidant and pharmacological properties of the extracts, with implications for their therapeutic applications.

### 3.1. Anti-Inflammatory Activity

The release of lysosomal enzymes during inflammation is implicated in the pathogenesis of various disorders, with extracellular enzyme activity linked to the severity and duration of inflammatory processes. Nonsteroidal drugs have been shown to exert their anti-inflammatory effects by either inhibiting these lysosomal enzymes or by stabilizing the lysosomal membrane. Given the similarity between the lysosomal membrane and the membrane of human red blood cells (HRBC), this study aimed to assess the stability of the HRBC membrane as a predictor of anti-inflammatory activity for four different extracts.

Each extract was incubated separately with HRBC solution at concentrations of 100, 200, and 300 µg/mL. The lysosomal-mediated mechanism for erythrocyte defense against hemolysis is dependent on lysosomal enzymes to maintain the integrity of RBC membranes by degrading damaged proteins and lipids due to oxidative or chemical stress. While adult erythrocytes do not possess typical lysosomes, lysosomal-like activities assist in cellular protection through the elimination of oxidized elements, management of calcium homeostasis, and prevention of membrane instability. Lysosomal membrane stabilization prevents harmful enzymes from escaping, potentially exacerbating hemolysis. The degree of hemolysis, indicative of membrane stability, was then compared with that induced by the standard drug, Diclofenac sodium, at equivalent concentrations. Remarkably, the methanol and ethyl acetate extracts at a concentration of 300 µg/mL demonstrated significant protection against hemolysis (*p* < 0.001) compared with the control. This observed protective effect may be attributed to the high phenolic content present in these two extracts, as outlined in Table 2. These findings underscore the potential anti-inflammatory properties of the methanol and ethyl acetate extracts, warranting further investigation into their therapeutic potential.

### 3.2. PPG Alleviated Acetic Acid Induced Injury in NCM460 Cells

#### Effect of PPG on NCM460 Cell Viability and Inflammatory Signaling Pathways

The NCM460 cell model was utilized to investigate the potential anti-inflammatory effects of PPG in vitro. Following treatment with acetic acid, a significant reduction in NCM460 cell viability was observed (Figure 2). However, intervention with PPG at doses ranging from 25 to 200 μg/mL maintained NCM460 cell viability above 90% for both 12 h and 24 h durations. Notably, pre-treatment with PPG conferred protection against acetic acid-induced cellular damage, indicative of its cytoprotective effects on NCM460 cells. Furthermore, the protective effect of PPG was found to be dose-dependent (Figure 2A,B), and PPG demonstrated inhibition of the NF-κB signaling pathway in vitro (Figure 2C–F). Subsequent investigation of Nuclear Factor erythroid 2-Related Factor 2 (Nrf2) expression in NCM460 cells revealed that high doses of PPG effectively upregulated Nrf2 expression and downregulated Kelch-like ECH-associated protein-1 (Keap-1) expression. These findings collectively demonstrate the ability of PPG to mitigate acetic acid-induced cellular damage in vitro.

To ascertain whether the observed mechanisms of PPG in inhibiting intestinal inflammation could be recapitulated in human cells, NCM460 cells were selected for further investigation. Treatment with acetic acid-induced extensive damage to NCM460 cells, including inhibition of cell growth and activation of the NF-κB pathway. Conversely, PPG treatment at a safe dose exhibited protective effects and attenuated the degree of damage, with inhibition of the NF-κB signaling pathway observed only at relatively high doses. Moreover, activation of the Nrf2 signaling pathway was evident only with 150 μg/mL PPG treatment, as evidenced by increased Nrf2 expression and decreased Keap-1 expression. Similar Nrf2 activation was observed with the treatment of NCM460 cells with PPG alone. These results indicate the potential of PPG as an anti-inflammatory and antioxidant agent, offering protection to epithelial cells.

Nrf2, which competes with NF-κB, plays a crucial role in modulating inflammatory signaling and oxidative stress. Activated Nrf2 inhibits NF-κB nuclear translocation, thereby attenuating inflammatory responses while exerting antioxidant effects. In vitro findings demonstrated that PPG activated the Nrf2 signaling pathway, contributing to its anti-inflammatory and antioxidant properties, which are potentially beneficial in the treatment of inflammatory bowel disease (IBD).

### 3.3. Acute Toxicity Studies In Vivo

The rat is exposed to a dose of 3, 30, 100, and 250 mg/kg body weight survived for more than three weeks following the initial exposure to the sample. The observations, including effects on respiration and sedation, are recorded in Table 3. Based on these findings, the sample was deemed safe up to a dosage of 100 mg/kg body weight. Dosages of 3, 30, and 100 mg/kg body weight were selected for further pharmacological investigation.

### 3.4. Recording of Body Weight of Animals from Various Groups

In the non-colitis control group, naïve animals exhibited a significant increase in body weight by day 8 (*p* < 0.001) and continued to demonstrate this increase by day 16 (*p* < 0.001). Conversely, animals in the colitis control group experienced a significant reduction in body weight by day 8 (*p* < 0.001). Following treatment with PPPG (at both doses), there was a gradual and significant increase in body weight observed. Additionally, the colitis control group showed a gradual reduction in average feed intake, while animals with colitis treated with PPPG exhibited a gradual increase in feed intake. These findings are summarized in Table 2 and illustrated in Figure 1.

### 3.5. Change in the Level of Malondialdehyde (MDA) in the Inflamed Colon of Experimental Animals

Colitis control recorded a significant increase in MDA concentration, and on the contrary, PPPG treatment in both selected low-dose and high-dose treated animals did not record any significant change in MDA concentration and was found to be of the same concentration (Figure 2A), (Table 3).

### 3.6. Change in the Level of Myeloperoxidase (MPO) in the Inflamed Colon of Experimental Animals

The colitis control group of animals recorded a significant increase in MPO concentration compared with the non-colitis control, and treatment with 3 mg/kg (low dose) did not significantly reduce the MPO concentration, and 30 mg/kg and 100 mg/kg treated animals recorded a significant (*p* > 0.001) reduction compared with the colitis control animals in a dose-dependent manner (Figure 2B, Table 3).

### 3.7. Ulcer Index of the Inflamed Colon of Experimental Animals from Various Groups

#### Colitis Control Group of Animals Recorded a Significant Increase in Ulcer Index

Colitic animals treated with a low dose (3 mg/kg) did not record a significant reduction in ulcer index, and on the contrary, colitic animals treated with medium and high doses, that is, 30 mg/kg and 100 mg/kg, respectively, recorded a significant reduction in the ulcer index. This effect is dose-dependent (Figure 3A), (Table 3).

### 3.8. Gross Morphological Changes in the Parts of GIT (Colon) Experimental Animals

Post euthanasia and dissection of the colon, the gross morphological changes observed were that the normal, non-colitis group showed a normal colon without any inflammation; the colitis control group showed significant damage, and the colon was heavily inflamed; and the treatment group colons showed less inflammation and signs of recovery.

### 3.9. Histopathological Studies

Histopathological data revealed the following findings groupwise.

Normal large intestine showing the simple columnar epithelial cells lining and a larger number of goblet cells (asterisk). The submucosal layer shows mild infiltration of mononuclear cells, mainly lymphocytes (arrow), and lamina propria with smooth muscle fibers; scale bar = 100 μm (Figure 3G(A)).

For colitis control, the large intestine shows mild damage to the normal villous architecture and lining epithelial cells with tissue debris in the lumen and loss of lining simple columnar epithelial cells with hyperplasia of goblet cells (asterisk). The submucosal layer shows mild infiltration of mononuclear cells, mainly lymphocytes (arrow); scale bar = 100 μm (Figure 3G(B)).

For the low-dose treatment group, the intestine shows intact cells and goblet cell hyperplasia (arrow) with mild infiltration of inflammatory cells. The intact lamina propria is seen in the submucosa with severe congestion of blood vessels (asterisk) and surrounding smooth muscle layers; scale bar = 100 μm (Figure 3G(C)). Histopathology findings of the intestine showing intact lining, simple columnar epithelial cells, and mild goblet cell hyperplasia (arrow) in a few areas. The intact lamina propria is seen in the submucosa with mild congestion of blood vessels (asterisk) and surrounding smooth muscle layers; scale bar = 100 μm (Figure 3G(D)).

## 4. Discussion

This study investigated the anti-inflammatory and therapeutic efficacy of PPPG in vitro and in vivo. The study focused on inflammatory pathway modulation and its impact on the major biochemical and histological markers of IBD. The well-established and proven method of acetic acid-induced colitis was followed in the study due to its resemblance to that of human UC in terms of superficial mucosal inflammation and the development of ulcers. The PPPG showed a very prominent dose-dependent activity.

Macroscopic scoring and histological observations indicate the previously reported protective effect of polyphenolic compound-rich crude drugs in alleviating intestinal inflammation. Apart from the macroscopical scoring, the study also incorporated histopathological findings of epithelial integrity, crypt architecture, and inflammatory infiltration that give additional insight into the tissue-level protective effect given by PPPG.

MPO levels were significantly reduced in moderate- and high-dose groups, indicating a lower neutrophil infiltration and reduction in acute inflammation. MDA levels did not show any significant lowering; being a marker for lipid peroxidation and oxidative stress, this supports the fact that PPPG shows immunomodulation but has no significant effect on lipid peroxidation and reactive oxygen species modulation. The effect might come with a higher dose, combination therapy, or prolonged exposure that can be taken as a scope for any future studies.

The extract showed high cytoprotection against acetic acid-induced cell injury in NCM460 colonic epithelial cells; key pro-inflammatory markers were also downregulated, fortifying the support of the immunomodulation pathway of the extract at the molecular mechanism level.

Overall, the study demonstrates the dose-dependent protective effect of peels, a part of the fruit that is usually thrown away, and opens the doors to newer research in the future that can explore various formulation strategies and delivery systems along with combination therapies using the same peel, therefore creating a cost-effective, socio-environmentally and economically viable alternative for the expensive biologic treatments. In the future, specific isolates and pharmacokinetics should also be assessed.

## 5. Conclusions

In conclusion, the findings of our study suggest that *Punica granatum* peel powder (PPPG) exhibits potential as a protective agent against acute colitis. Through oral acute toxicity testing, we determined that PPPG is safe up to a dose of 250 mg/kg, with no observed signs of mortality or morbidity during the observation period except mild lacrimation and loose stool. The dose showed no mortality up to 2000 mg/kg as well. The selected doses were 3 mg/kg, 30 mg/kg, and 100 mg/kg, which span two orders of magnitude that can be ideal for an exploratory study of effect.

Our investigation focused on evaluating the dose-dependent effects of oral PPPG administration on colitic animals, assessing changes in body weight, feed intake, ulcer index, myeloperoxidase (MPO) and malondialdehyde (MDA) levels, biochemical markers of neutrophil migration, gross colon changes, and histopathological evaluation. Treatment with PPPG resulted in significant weight gain and increased food consumption in test animals, indicating a reversal of acute colitis. Additionally, PPPG-treated animals exhibited reduced ulcer index scores, particularly at the higher dose of 1000 mg/kg. However, PPPG treatment did not fully protect against oxidative stress, as evidenced by MDA levels. Notably, higher doses of PPPG were associated with reduced neutrophil migration, a key early inflammatory response, whereas lower doses did not exhibit this effect. Histopathological analysis revealed mild inflammatory cell infiltration and less severe congestion in the bowel vessels of higher-dose-treated animals compared with lower-dose-treated ones. Overall, our results suggest that PPPG, particularly at higher doses, exerts a significant protective effect against acetic acid-induced colitis. Further investigation into the specific chemical constituents responsible for this activity is warranted, as it may pave the way for the development of PPPG as a prophylactic intestinal protective agent. Our study contributes valuable insights into the potential therapeutic applications of herbal-based therapies for chronic inflammatory conditions, emphasizing the role of PPPG in modulating the IL-6/STAT3 pathway and demonstrating its efficacy in a preclinical model of colitis.

## Figures and Tables

**Figure 1 pharmaceutics-17-00843-f001:**
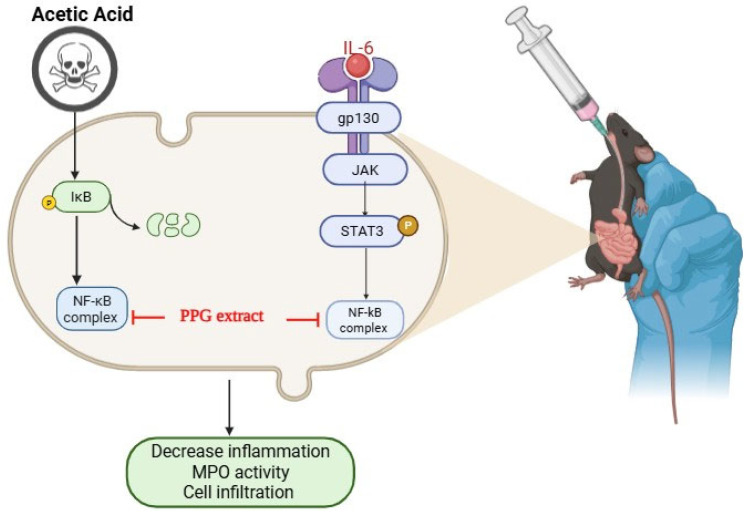
Schematic representation of Acetic acid induces IBD management by PPG extract.

**Figure 2 pharmaceutics-17-00843-f002:**
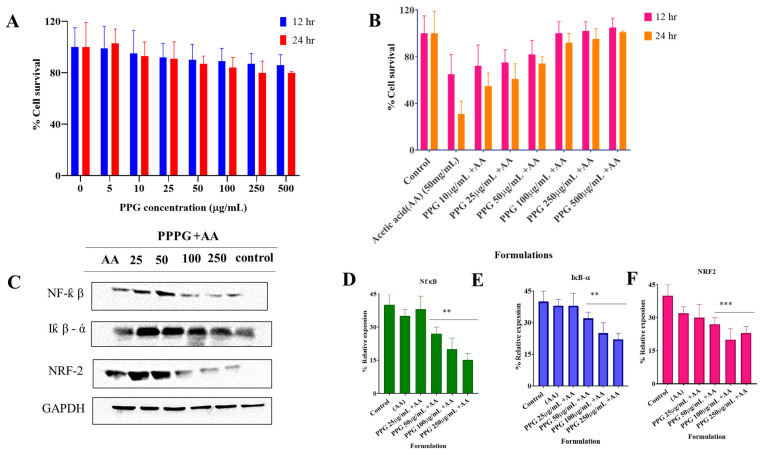
PPPG significantly alleviated acetic acid-induced injury in NCM460 cells. (**A**,**B**) Treatment with PPPG significantly restored cell viability reduced by acetic acid (AA) exposure. Cell viability was measured using MTT assay, and results are expressed as mean ± SD (*n* = 5). (**C**–**F**) Western blot analysis revealed that PPPG significantly inhibited the expression of phosphorylated NF-κB and IκB-α while upregulating the expression of Nrf2 and downregulating Keap-1. Densitometric quantification of protein bands was normalized to β-actin and analyzed via ImageJ software. Statistical analysis confirmed significant differences across treatment groups (** *p* < 0.05 to *** *p* < 0.001).

**Figure 3 pharmaceutics-17-00843-f003:**
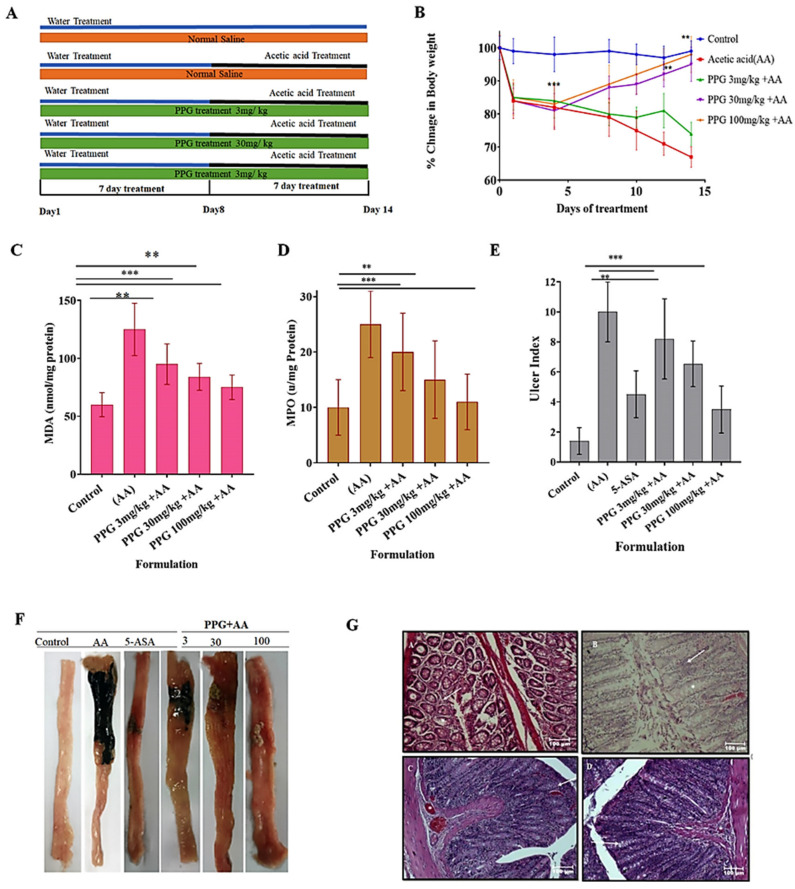
PPG alleviated AA-induced colitis symptoms in rat (**A**). Experimental design. (**B**). Changes in body weight. (**C**). Status of malondialdehyde (MDA) levels in the colons of the control, acetic acid (AA), AA  +  PPG (3, 30, and 100 mg/kg) ** *p* < 0.01 and *** *p*  <  0.001, compared with the control group. Data are expressed as means  ±  SEM. (n = 5). The statistical analysis was carried out by one-way analysis of variance (ANOVA) followed by Tukey’s post hoc test. (**D**). Effect of PPG extract (3, 30, and 100 mg/kg) on the level of myeloperoxidase (MPO) activity in the colon tissue. Results are expressed as mean ± SEM (*n* = 5). ** *p* < 0.01 and *** *p* < 0.001 compared with control group. The statistical analysis was carried out by one-way analysis of variance (ANOVA) followed by Tukey’s post hoc test. (**E**). Effect of 5-ASA (2 mg/kg) and PPG extract (3, 30, 100 mg/kg) on ulcer index. Data are expressed as mean ± SEM (*n* = 5). ** *p* < 0.01 and *** *p* < 0.001 compared with control group. The statistical analysis was carried out by one-way analysis of variance (ANOVA) followed by Tukey’s post hoc test (**F**). Images of colon tissue 14 days after colitis induction. control; acetic acid (AA); 5-ASA and different concentrations of PPG extract test (**G**). Histopathological images of different groups. Non-colitis (A), colitis control (B), low-dose treatment (C), high-dose treatment (D).

**Table 1 pharmaceutics-17-00843-t001:** Phytochemical screening of various extracts of fruit peel of *P. granatum* L.

Constituents	Water	Ethanol	Methanol	Ethyl Acetate
Carbohydrate	+	+	+	−
Phytosterol	−	−	−	−
Fixed oil	−	−	−	−
Alkaloid	−	−	−	−
Glycoside	−	+	+	−
Saponin	+	−	−	−
Flavonoid	−	−	+	+
Tannin		+	+	+

(+) Present, (−) Absent.

**Table 2 pharmaceutics-17-00843-t002:** Anti-inflammatory activity of the Peel extract on HRBC membrane stabilization method.

Treatment	100 µg/mL	200 µg/mL	300 µg/mL
Control (Distilled Water)	100.00 ± 0.00 (0.00%)	–	–
Water Extract	46.76 ± 2.03 (53.24%)	41.18 ± 1.06 (58.82%)	36.26 ± 2.05 (63.74%)
Ethanolic Extract	39.12 ± 1.02 (60.88%)	34.55 ± 1.19 (65.45%)	30.76 ± 0.06 (69.24%)
Methanolic Extract	08.75 ± 0.03 (91.25%) *	05.16 ± 1.11 (94.84%) *	00.74 ± 2.04 (99.26%) *
Ethyl Acetate Extract	12.27 ± 0.62 (87.73%) *	04.26 ± 0.98 (95.74%) *	02.87 ± 1.54 (97.13%) *
Diclofenac Sodium	02.87 ± 1.20 (97.13%) *	02.38 ± 0.90 (97.62%) *	00.97 ± 0.54 (99.03%) *

The data represented as mean ± SD, *n* = 5 Values outside brackets are % hemolysis; values in brackets are % protection. * Significant difference compared with control (*p* < 0.001).

**Table 3 pharmaceutics-17-00843-t003:** Observation of different parameters after a single high oral dose of peel powder of *Punica granatum*.

Dose (mg/kg)	Skin Color	Diarrhea	Lacrimation	Sedation	Respiration
3	No change	Not observed	No	No	Normal
30	No change	Not observed	No	No	Normal
100	No change	Not observed	No	No	Normal
250	No change	Little loose tools (20% animals)	Frequently (50% animals)	No	Abnormal

## Data Availability

Data will be Provided on Request to the Corresponding Author.

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
