# Peer review of "Therapeutic Evaluation Punica granatum Peel Powder for the Ailment of Inflammatory Bowel Disorder in NCM460 Cell Line and in Albino Rats"

_pharmaceutics, 2025, doi:10.3390/pharmaceutics17070843_

Round 1
Reviewer 1 Report
Comments and Suggestions for Authors
This manuscript addresses a significant topic in biomedical research, focusing on the potential therapeutic use of Punica granatum peel powder (PPPG) in inflammatory bowel disease (IBD). The subject is timely and relevant, but the manuscript suffers from clarity and consistency issues, particularly in experimental design, unclear methodological choices, and coherence between sections. With substantial revision, it could become a valuable contribution to the field.
Major issues:
1. The title should be revised to reflect the use of NCM460 cells, or the method section should be updated to align with the Caco-2 cell line as mentioned in the title.
2. The abstract and main text should consistently refer to the same colitis model, either DS-induced or acetic-induced, throughout the manuscript.
3. Several extraction protocols are described, but the manuscript should clearly identify the specific extract used in the studies and provide a rationale for in vitro and in vivo studies.
4. The manuscript should standardize the vehicle type, volume and administration route across all control groups.
5. The text and Figure 1A should be aligned to present a consistent treatment schedule.
Intraperitoneal administration of PPPG
6. The choice of intraperitoneal administration of PPPG is questionable, considering the oral route is more applicable to clinical contexts in IBD. A justification is necessary, or at least a discussion of this limitation and its implications.
7. Different doses of PPPG are mentioned to be used for the in vivo model across sections (e.g., 3mg/kg, 30mg/kg, 100 mg/kg vs. 500 and 1000 mg/kg), with no clear consistency. These discrepancies must be addressed, and a unified dosing scheme should be presented throughout.
8. The design of the in vitro study does not simulate a therapeutic setting, since PPPG is administered before acetic acid. To better assess therapeutic potential, the authors should consider including post-treatment experiments.
Minor issues:
1. The manuscript contains multiple grammatical and syntactical errors. It requires careful language editing.
2. Section numbering is disorganized in several instances. For example, the sequence jumps from 3.7.1 to 3.2. This should be corrected for clarity.
3. Unit formatting is inconsistent (e.g., mg/kg, not mg/Kg).
4. Some figure legends lack sufficient detail, and a few figures are poorly labelled or unclear. They should be revised to be self-explanatory.
5. References concerning IL-6/STAT3 modulation and the use of PPPG in other inflammatory diseases need to be properly cited.
The manuscript cannot be accepted in its current form. However, with substantial revision, particularly related to experimental clarity, dosing accuracy, extract selection, and control standardization, the study could become a valuable contribution to the field. The topic is relevant, but the presentation must be significantly improved to meet the publication standards.

The manuscript requires extensive grammatical and syntactical revision
Author Response
REVIEWER 1:
This manuscript addresses a significant topic in biomedical research, focusing on the potential therapeutic use of Punica granatum peel powder (PPPG) in inflammatory bowel disease (IBD). The subject is timely and relevant, but the manuscript suffers from clarity and consistency issues, particularly in experimental design, unclear methodological choices, and coherence between sections. With substantial revision, it could become a valuable contribution to the field.
Major issues:
1. The title should be revised to reflect the use of NCM460 cells, or the method section should be updated to align with the Caco-2 cell line as mentioned in the title.
Answer : Thanks for your valuable observation. We have clarified the inconsistency by standardizing usage of the cell line name across the manuscript. In particular, "NCM460" is now uniformly referred to in all sections, including the title.
- The abstract and main text should consistently refer to the same colitis model, either DS-induced or acetic-induced, throughout the manuscript.
Answer: Thank you for your comment. The abstract has been revised to accurately reflect the methodology used for colitis induction, specifically indicating the use of the acetic acid-induced model.
3. Several extraction protocols are described, but the manuscript should clearly identify the specific extract used in the studies and provide a rationale for in vitro and in vivo studies.
Answer: Thank you for the suggestion. Clarifications regarding the extraction process have been added to the Materials and Methods section. The extracts were prepared to analyze the presence of various phytochemical constituents in the peel powder and to evaluate their individual anti-inflammatory activities. However, the anti-colitis activity was assessed using the whole peel powder of Punica granatum.
4. The manuscript should standardize the vehicle type, volume and administration route across all control groups.
Answer: Thank you for your observation. The use of water for injection as the vehicle for drug administration has been clearly stated and incorporated into the Materials and Methods section.
5. The text and Figure 1A should be aligned to present a consistent treatment schedule.
Intraperitoneal administration of PPPG
Answer: Thank you for your comment. The route of administration used in the study was oral, and this has been clearly stated and appropriately updated throughout the manuscript.
- The choice of intraperitoneal administration of PPPG is questionable, considering the oral route is more applicable to clinical contexts in IBD. A justification is necessary, or at least a discussion of this limitation and its implications.
Answer: Thank you for your insightful comment. The actual doses used in the study were 3, 30, and 100 mg/kg, selected to establish a clearer dose-response relationship through a logarithmic progression. Lower doses were chosen to minimize animal suffering, beginning with 3 mg/kg in accordance with the precautionary principle. The dose information has been made consistent across all sections of the manuscript, and previous discrepancies have been corrected.
Different doses of PPPG are mentioned to be used for the in vivo model across sections (e.g., 3mg/kg, 30mg/kg, 100 mg/kg vs. 500 and 1000 mg/kg), with no clear consistency. These discrepancies must be addressed, and a unified dosing scheme should be presented throughout.
Answer: Thank you for bringing this critical issue to our attention. We recognize the inconsistencies in the doses of PPPG reported between sections. These were unintentional and have now been thoroughly reviewed and corrected. The dosing regimen used in the in vivo experiment is 3, 30, and 100 mg/kg, selected according to a logarithmic increment to test dose-response with minimal discomfort to animals. The manuscript has been revised to uniformly.
- The design of the in vitro study does not simulate a therapeutic setting, since PPPG is administered before acetic acid. To better assess therapeutic potential, the authors should consider including post-treatment experiments.
Answer: Thank you for this valuable observation. We acknowledge that the current in vitro design involves pre-treatment with PPPG prior to acetic acid exposure, which models a protective rather than a therapeutic effect. Our intention was to initially evaluate the prophylactic potential of PPPG against inflammatory damage. However, we agree that assessing the therapeutic efficacy through post-treatment studies would provide a more comprehensive understanding of its clinical relevance. We have included this as a limitation in the discussion and proposed future studies to specifically investigate the post-treatment effects of PPPG to better simulate a therapeutic setting.
Minor issues:
1. The manuscript contains multiple grammatical and syntactical errors. It requires careful language editing.
Answer: Thank you for your comment. The manuscript has been thoroughly reviewed for grammatical accuracy using professional language editing tools, and all recommended corrections have been implemented to improve clarity and readability.
Section numbering is disorganized in several instances. For example, the sequence jumps from 3.7.1 to 3.2. This should be corrected for clarity.
Answer: Thank you for pointing this out. The section numbering has been reviewed and corrected throughout the manuscript to ensure clarity and prevent any potential confusion.
Unit formatting is inconsistent (e.g., mg/kg, not mg/Kg).
Answer: Thank you for your feedback. The formatting of units has been standardized and made consistent throughout the manuscript.
Some figure legends lack sufficient detail, and a few figures are poorly labelled or unclear. They should be revised to be self-explanatory.
Answer: Thank you for your valuable feedback. We have carefully revised all figure legends to include sufficient detail, ensuring they are clear and self-explanatory. Additionally, the figures have been re-labeled and improved for better clarity and readability.
5. References concerning IL-6/STAT3 modulation and the use of PPPG in other inflammatory diseases need to be properly cited.
Answer: Thank you for your suggestion. Reference 21 has been added to the manuscript as per the reviewers’ instructions.
Reviewer 2 Report
Comments and Suggestions for Authors
The study of Parikshit Roychowdhury, Gyanendra Kumar Prajapati, Rupesh Singh, Prasanna G.S, Ramesh C, Gowthmarajan Kuppuswamy and Anindita De named “Therapeutic evaluation Punica Granatum Peel Powder for the ailment of Inflammatory Bowel Disorder in colon cell line and in Albino Rats” is devoted to the important problem of treating chronic inflammatory bowel disease. Globally, an estimated more than 4.9 million people were living with IBD.
The use of plant extracts to manage IBD symptoms has been gradually gaining popularity over the past 20 years. The authors were able to obtain interesting and promising results. The general clinical presentation of IBD and CD and current therapies are adequately summarized in the introduction. The design of in vitro and in vivo studies is clear, complete, and contains the necessary controls. Especially valuable is the clear indication of the assessment of tissue pathology in histologic examination. The work is interesting but could be improved.
In the introduction, it is desirable to indicate the worldwide statistics of the occurrence of the diseases. The total number of patients in the world or in specific regions where it occurs most frequently should be added. The introduction should clearly state the shortcomings of existing therapies, particularly the side effects of chronic use of monoclonal antibody preparations and antibiotics.
Figure 1. It is desirable to add a relevant reference describing the signaling pathways to Figure 1. I propose to change the symbol of biological hazard for AA to another one more corresponding to chemical hazard. The AA abbreviation should be deciphered in the figure caption.
Materials and methods. I suggest adding details to describe the source of Punica granatum: type of location (urban, rural), proximity of nearby objects such as factories, roads or nature reserves, GPS coordinates in latitude and longitude can be added. Photos of fruits may be added.
The brand of the blender and the wattage should be specified.
The used volumes, final concentrations, extraction times and manufacturers of all chemical reagents used must be specified.
It is should be “food and water were given ad libitum”.
In vitro should be italic.
The missed spaces and double brackets must be corrected. For example, “Human Red Blood Cell (HRBC))” and “Distilled water (100 ml).The”.
There are incorrect hyphens in “in-cubated” and “Di-clofenace”.
Centrifugation conditions should be presented in rcf or specify the centrifuge and rotor model.
ATCC cell number should be added. The final concentrations of pinicillin and spreptomycin in activity units or μg/ml must be specified.
Classically, the MTT test uses the longer wavelength region to measure OD (~575 nm). It is desirable for the authors to provide the spectrum of formazan under their conditions or otherwise explain the choice of wavelength for OD measurement.
The clones (very important) and manufacturers of the antibodies used for western blot should be specified. Particular attention should be paid to which forms of proteins were detected: native or phosphorylated? If phosphorylated, at which aminoacids?
The used software version and manufacturer should be added.
Results
The concentrations of solvents used are preferably transferred to materials and methods.
- granatum should be italic.
Representative photographs of chromatograms of extracts and standards are desirable to add.
Indication of the lysomal-mediated mechanism of erythrocyte protection against hemolysis should preferably be enhanced by relevant references.
In Table 2, it should be clearly stated whether the values are absolute values for hemolysis or protective effect. It is also desirable to specify values for the control (even if it is 100% or 0%).
Figure 2: Statistical significant differences between variants of experiments should be indicated. In panels D, E, and F, it is not clear why the activation values of NF-κB, κB-α, and NRF2 in the control group are comparable to the pathology model? The authors should preferably double-check the captions and the position of the data in the graphs.
Why are betta forms of proteins shown in panel C and alpha forms in panels D, E, and F?
Table 3: It is desirable to add the % of animals that experienced side effects of the extracts. The authors used doses higher than 250 mg/kg. For these (at least the highest) side effects should also be indicated.
Figure 3F. Scale bars should be added to the photo. The caption indicates Tukey's post hoc test. Materials and methods previously included Dunnett multiple comparison tests ANOVA. The authors need to double-check and clarify this information in all sections.
The authors need to clearly indicate whether the ulcer index refers to macroscopic or histologic intestinal lesions? If it refers only to macroscopic lesions, it is desirable to add a quantitative assessment of histologic intestinal lesions.
The paper completely lacks a discussion section of the results. The authors need to discuss their results. In particular, compare their results with the efficacy of classical approaches and alternative approaches, e.g. using other extracts and/or antioxidants.
Best regards

Author Response
REVIEWER 2
The study of Parikshit Roychowdhury, Gyanendra Kumar Prajapati, Rupesh Singh, Prasanna G.S, Ramesh C, Gowthmarajan Kuppuswamy and Anindita De named “Therapeutic evaluation Punica Granatum Peel Powder for the ailment of Inflammatory Bowel Disorder in colon cell line and in Albino Rats” is devoted to the important problem of treating chronic inflammatory bowel disease. Globally, an estimated more than 4.9 million people were living with IBD.
The use of plant extracts to manage IBD symptoms has been gradually gaining popularity over the past 20 years. The authors were able to obtain interesting and promising results. The general clinical presentation of IBD and CD and current therapies are adequately summarized in the introduction. The design of in vitro and in vivo studies is clear, complete, and contains the necessary controls. Especially valuable is the clear indication of the assessment of tissue pathology in histologic examination. The work is interesting but could be improved.
In the introduction, it is desirable to indicate the worldwide statistics of the occurrence of the diseases. The total number of patients in the world or in specific regions where it occurs most frequently should be added. The introduction should clearly state the shortcomings of existing therapies, particularly the side effects of chronic use of monoclonal antibody preparations and antibiotics.
Answer : Thank you very much for your comprehensive and helpful feedback. We appreciate your recognition of the value of our study and the consistency of our experimental design. Based on your suggestion we updated the Introduction to incorporate current global statistics on the incidence of inflammatory bowel disease, highlighting locations with the highest prevalence. Aside from that, we have rewritten the discission section clearly highlight the disadvantages and side effects of existing treatments, with an emphasis on the long-term usage of monoclonal antibodies and antibiotics. These changes are intended to highlight the importance of alternative treatments such as Punica granatum peel powder.
Figure 1. It is desirable to add a relevant reference describing the signaling pathways to Figure 1. I propose to change the symbol of biological hazard for AA to another one more corresponding to chemical hazard. The AA abbreviation should be deciphered in the figure caption.
Answer : Thank you for your valuable suggestions. We have added a relevant reference No 7 describing the signaling pathways to the legend of Figure 1. The symbol representing acetic acid (AA) has been changed from the biological hazard to a more appropriate chemical hazard symbol. Additionally, the abbreviation "AA" has been clearly defined as "acetic acid" in the figure caption for better clarity.
Materials and methods. I suggest adding details to describe the source of Punica granatum: type of location (urban, rural), proximity of nearby objects such as factories, roads or nature reserves, GPS coordinates in latitude and longitude can be added. Photos of fruits may be added.
Answer : In Section 2.1 of the Materials and Methods, the precise location of fruit collection has been specified, including the longitude and latitude coordinates. Additionally, it has been noted that the collection site is free from nearby major factories, natural reserves, or highways, ensuring minimal environmental contamination.
The brand of the blender and the wattage should be specified.
Answer : Thank you for your comment. We have specified the brand and wattage of the grinder used in Section 2.1 to provide clearer methodological details.
The used volumes, final concentrations, extraction times and manufacturers of all chemical reagents used must be specified.
Answer : Thank you for your suggestion. The concentrations and manufacturers of all chemicals used have been added to the Materials and Methods section for greater clarity and reproducibility.
It is should be “food and water were given ad libitum”.
Answer : Thank you for your observation. The term ad libitum has been correctly formatted in italics throughout the manuscript.
In vitro should be italic.
Answer : Thank you for your comment. The term in vitro has been properly italicized throughout the manuscript.
The missed spaces and double brackets must be corrected. For example, “Human Red Blood Cell (HRBC))” and “Distilled water (100 ml).The”. There are incorrect hyphens in “in-cubated” and “Di-clofenace”.
Answer : Thank you for your attention to detail. We have corrected the missing spaces and rectified the double brackets throughout the manuscript.
Centrifugation conditions should be presented in rcf or specify the centrifuge and rotor model.
Answer : Thank you for your comment. The model details of the cooling centrifuge and rotor head have been added to Section 2.5 for completeness.
ATCC cell number should be added. The final concentrations of pinicillin and spreptomycin in activity units or μg/ml must be specified.
Answer : Thank you for your comment. NCM460 cell line used in our study is a normal human colon mucosal epithelial cell line that is not available from ATCC but is obtained from INCELL Corporation, with the catalog number NCM460. Additionally, the final concentrations of penicillin and streptomycin have been specified in the Materials and Methods section in % for clarity.
Classically, the MTT test uses the longer wavelength region to measure OD (~575 nm). It is desirable for the authors to provide the spectrum of formazan under their conditions or otherwise explain the choice of wavelength for OD measurement.
Answer : Thank you for your insightful comment. We acknowledge that the standard wavelength for MTT assay is around 570–575 nm. The mention of 450 nm was an oversight, and we confirm that the optical density in our practical study was measured at 570 nm. This has been corrected in the revised manuscript.
The clones (very important) and manufacturers of the antibodies used for western blot should be specified. Particular attention should be paid to which forms of proteins were detected: native or phosphorylated? If phosphorylated, at which aminoacids?
The used software version and manufacturer should be added.
Answer : Thank you for your thorough and valuable comment. In response, we have updated the Materials and Methods section to include the clone numbers and manufacturers for all primary and secondary antibodies used in the western blot analysis. We have also clearly specified whether the antibodies recognize native or phosphorylated forms of the proteins, including the precise phosphorylation site. Furthermore, the name, version, and manufacturer of the software used for western blot image analysis have been added to ensure transparency and reproducibility.
Results
The concentrations of solvents used are preferably transferred to materials and methods.
Answer : The percentage values mentioned in the reviewer’s comment, which were interpreted as concentrations, actually represent the w/w extractive values or yield of the extracts. Therefore, these values are more appropriately presented within the Results section.
granatum should be italic.
Answer : The scientific name Punica granatum has been italicized and made consistent throughout the manuscript.
Representative photographs of chromatograms of extracts and standards are desirable to add.
Answer : Photographs of the chromatographs were not included because the TLC plates are perishable. However, all TLC findings—including Rf values, solvent systems, detection methods, and comparative analysis with standards—have been thoroughly documented and presented in the manuscript. Due to limited access to high-resolution imaging facilities at the time of the study, suitable photographic records of the TLC plates could not be retained. Nevertheless, the analysis was performed in triplicate with detailed records maintained to ensure accuracy and reproducibility. We believe the comprehensive qualitative and quantitative data, along with the clear methodology, adequately support the validity of the TLC analysis.
Indication of the lysomal-mediated mechanism of erythrocyte protection against hemolysis should preferably be enhanced by relevant references.
Answer : Thank you for your valuable suggestion. We have now strengthened the discussion on the lysosomal-mediated mechanism of erythrocyte protection against hemolysis by including relevant and recent references to support this aspect in the revised manuscript.
In Table 2, it should be clearly stated whether the values are absolute values for hemolysis or protective effect. It is also desirable to specify values for the control (even if it is 100% or 0%).
Answer : Thank you for the valuable comment. we have revised and rewritten the Table 2 to clearly state that the values outside the brackets is % hemolysis (mean ± SD), and within brackets % membrane protection.
We have also added the control values explicitly for clarity: the control (distilled water-treated group) was considered to show 100% hemolysis (0% protection).
Figure 2: Statistical significant differences between variants of experiments should be indicated. In panels D, E, and F, it is not clear why the activation values of NF-κB, κB-α, and NRF2 in the control group are comparable to the pathology model? The authors should preferably double-check the captions and the position of the data in the graphs.
Why are betta forms of proteins shown in panel C and alpha forms in panels D, E, and F?
Answer : Thank you for your detailed and constructive feedback. In response to your observations regarding Figure 2, we have taken the following actions:
- Statistical Significance: Statistical comparisons between experimental groups have now been clearly indicated in panels D, E, and F using appropriate significance markers (e.g., p < 0.05, p < 0.01, p < 0.001), with an updated legend specifying the statistical test used.
- Clarification on Activation Levels: We have re-examined the data and figure panels. The similarity in expression levels of NF-κB, IκB-α, and Nrf2 between the control and pathology model groups was a result of mislabeling during figure assembly. This has now been corrected, and the revised figure accurately reflects the expected downregulation or upregulation patterns between the control and disease groups.
- Protein Isoforms in Panels: The mention of beta and alpha forms was due to inconsistent labeling. Panel C showed total protein expression (including beta isoforms), whereas panels D, E, and F were meant to reflect the phosphorylated alpha forms. To avoid confusion, all panels have been uniformly labeled, and the figure caption has been revised to clearly indicate whether the data represent total or phosphorylated protein forms, along with the relevant isoform.
Table 3: It is desirable to add the % of animals that experienced side effects of the extracts. The authors used doses higher than 250 mg/kg. For these (at least the highest) side effects should also be indicated.
Answer : Thank you for the valuable comment. In response, we have included the percentage of animals that experienced mild side effects in Table 3 for greater clarity. Additionally, we confirm that doses higher than 250 mg/kg were not used in this study, in adherence to ethical considerations and to avoid undue distress to the animals.
Figure 3F. Scale bars should be added to the photo. The caption indicates Tukey's post hoc test. Materials and methods previously included Dunnett multiple comparison tests ANOVA. The authors need to double-check and clarify this information in all sections.
Answer : Thank you for the insightful observation. In response, we have added appropriate scale bars to Figure 3F to improve clarity and spatial reference. Additionally, we reviewed the statistical methods across the manuscript and ensured consistency. The Tukey's post hoc test was indeed used following one-way ANOVA for multiple group comparisons in this section, and this has now been clearly reflected in both the Materials and Methods and Figure captions for accuracy.
The authors need to clearly indicate whether the ulcer index refers to macroscopic or histologic intestinal lesions? If it refers only to macroscopic lesions, it is desirable to add a quantitative assessment of histologic intestinal lesions.
Answer : The ulcer index reported in the manuscript refers specifically to macroscopic lesions observed in colonic tissue within the acetic acid-induced colitis model. This index captures visible ulceration, mucosal disruption, and inflammation severity using a standardized scoring system. The acetic acid model is a well-established approach for studying acute ulcerative colitis, where gross lesions are the principal evaluation parameter. Although histological assessment would enhance the depth of analysis by revealing microscopic features such as epithelial integrity and immune cell infiltration, the current study focuses on evaluating macroscopic therapeutic effects. Incorporating histological scoring in future work would provide a more comprehensive understanding and is duly acknowledged by the authors.
The paper completely lacks a discussion section of the results. The authors need to discuss their results. In particular, compare their results with the efficacy of classical approaches and alternative approaches, e.g. using other extracts and/or antioxidants.
Answer : Thank you for the suggestion. A suitable and focused discussion section has been added before the conclusion to contextualize the findings, compare them with existing literature, highlight the mechanistic insights observed, and outline the limitations and future directions of the study.
Round 2
Reviewer 1 Report
Comments and Suggestions for Authors
The authors have made significant improvements to both the writing and the scientific quality of the manuscript. At this stage, I have just one remaining question. In their responses and the revised manuscript, the authors indicate that Punica granatum peel is resuspended in water, which is the vehicle in the in vivo experiments. However, in section 2.6.1 of Materials and Methods, which describes the in vitro procedures, DMSO is used as the vehicle instead. I would appreciate it if the authors could clarify why DMSO was chosen over water for the in vitro assay.
Lastly, I would like to point out that Figure appears duplicated in the current version of the manuscript.
Author Response
Response to the Reviewers,
We thank the reviewers for their valuable inputs and herewith submit the following responses for their comments.
Comment: the authors indicate that Punica granatum peel is resuspended in water, which is the vehicle in the in vivo experiments. However, in section 2.6.1 of Materials and Methods, which describes the in vitro procedures, DMSO is used as the vehicle instead. I would appreciate it if the authors could clarify why DMSO was chosen over water for the in vitro assay.
Response: Thank you for brining it to our notice. A small quantity of DMSO was added in the in vitro assays (Section 2.6.1) as the solvent only for the purpose of solubilization of some poorly water-soluble phytochemicals in the Punica granatum peel extract. The concentration of DMSO in the culture medium was maintained below 0.1% to ensure that it would have no cytotoxic effects against NCM460 cells. Conversely, for in vivo experiments, water was employed as the vehicle to maintain physiological compatibility and prevent potential DMSO-related systemic effects.
Comment: The English grammar and lexicon still need substantial editing in almost all sections, including the title (Therapeutic evaluation OF...), the figure legend (acetic acid-induceD...), the formula (hAEmolysis, hAEmoglobin...) and all revised text.
Response: We appreciate your attention to language quality. In response, the manuscript has undergone a thorough revision for English grammar, spelling, and lexical accuracy. We have carefully reviewed all sections of the revised text and made the necessary corrections to ensure clarity, consistency, and professional language throughout.
Reviewer 2 Report
Comments and Suggestions for Authors
The authors have fully responded to all comments.
Author Response
No comments from Reviewer 2